# OpenReview forum: "SynerGPT: In-Context Learning for Personalized Drug Synergy Prediction and Drug Design"
_colmweb.org/COLM/2024/Conference — COLM_

### Official Review · Reviewer_XtiT · 2024-04-12

**Rating:** 5
**Confidence:** 4
**Ethics Flag:** 2

**Summary:**

This paper proposes to use transformer-based language models to solve the problem of drug synergy prediction. In particular, the few-shot DDS problem is studied with the technique of in-context learning. In addition, authors also study the task of drug retrieval with the proposed model.

**Ethics Concerns Details:**

Patients personalized data.

**Reasons To Accept:**

1. The idea of using LLM for drug synergy prediction and drug design without other resources is interesting.
2. The introduction is smooth and clear.
3. The design of synergy graph based context selection for few-shot DDS prediction is novel.

**Reasons To Reject:**

1. The main problem of this paper is the organization of methodology.
First, Section 4 seems unnecessary and redundant. a) There are several works in the literature showing that BERT can do DDI prediction, DDS prediction, and link prediction on graphs. b) The experiments in this part are on encoder-only model while the remaining parts are on decoder-only model. c) The main problem setup in this paper is for few-shot DDS, while the experiments in section 4 are not about this.
Second, it will be much better if sections 5-7 can be organized as a whole framework. Currently, the DDS and drug design parts are too separate, with different algorithms and different training strategies.
Third, the results are given for each section, again making the methodologies look separated.

2. The authors mentioned several times about patient's personalized dataset (also in the title). But how to understand the relationship between the proposed method and the work "personalized"? There is little description and motivation for this point in the method design. In addition, I also fail to find how such a dataset looks like. How many patients are there and how many samples each patient has?

3. It is hard for me to understand the results in Figure 2. In other words, how can I see that the retrieved molecules are meaningful? Why does the proposed method work better than other retrieval methods?

4. There lacks an important baseline/related work studying the few-shot DDS problem: "Few-Shot Drug Synergy Prediction With a Prior-Guided Hypernetwork Architecture" in TPAMI 2023.

Overall, even though this paper introduces something novel and interesting, there is a huge space to improve in this paper.

---

> ### Author Rebuttal · Authors · 2024-05-31
>
> Thank you for your review and recognition of the novelty of our approach.
>
> > Organization of the Methodology
> * We will take your comments into consideration for improving the clarity of the paper, such as by improving connections between sections 5-7 to make the paper more accessible.
>
> * Regarding a): Sec. 4 is actually quite different to existing work on BERT for DDI: it shows that language models such as BERT, without features, (i.e., using *random tokens* as input) are performant on the drug synergy task. This result is initially unintuitive, and it supports our motivation to use language models for in-context learning of drug synergies on unknown drugs/cells.
> Would you please be able to point out the work you are referring to?
>
> > Patient's personalized dataset
> * Thank you for pointing this out; we will add more details in the camera-ready. One of the settings we explore is “unknown cell lines”. These often come from a specific patient’s tumor cell biopsy (as in the benchmark datasets we use).
> * We discuss a “standardized clinical assay for drug synergy prediction” for creating a personalized dataset. It would work by:
> 1. A biopsy would be taken from a patient tumor.
> 2. The biopsy would be tested in the proposed assay against 10-20 drug combos (selected by §6). This would give us a patient-specific personalized dataset.
> 3. This dataset can be used for patient-specific drug synergy predictions via SynerGPT.
>
> > Figure 2 and retrieval methods
> * Our method presented in §7 is extending SynerGPT to a new task: inverse drug design via retrieval where the query is synergy tuples as input. Figure 2 shows an example; as more synergy tuples are provided to the model in-context, it is able to predict structures closer to the ground truth.
> * This is the first exploration of this novel task; other retrieval methods are not designed to work with drug synergy tuples as input, so they cannot be used to compare. However, we 1) used one of the strongest current representations for molecules, MegaMolBARTv2, and 2) compared multiple different losses, including data augmentation (Appendix F Figure 4 shows results).
>
> > TPAMI Paper
> * Thank you for pointing out this recent paper! We will cite and discuss it in our related work. We tested their method on our dataset splits and found it to have a ROC-AUC of 81.3 compared to 83.8 from SynerGPT (Table 2).
> * A recent review (arxiv.org/abs/2404.02484v2) found it to be outperformed by our baselines GraphSynergy and DeepDDS.

---

> > ### Comment · Reviewer_XtiT · 2024-06-06
> > **After rebuttal comments**
> >
> > Thank you for your responses. I have increased the score as my appreciation for the responses.
> >
> > The remaining concerns, as indicated by other reviewers, are the presentation issues and the limited experiments. Hope the authors will take these concerns into account and carefully revise the presentations if accepted.

---

### Official Review · Reviewer_FRwf · 2024-05-08

**Rating:** 7
**Confidence:** 4
**Ethics Flag:** 1

**Summary:**

The paper introduces SynerGPT, a novel machine learning framework that leverages a transformer model (specifically GPT) to predict drug synergies without the need for domain-specific data like molecular fingerprints or protein interactions. This approach is notable for its potential to simplify and accelerate the process of identifying effective drug combinations for cancer treatment, a crucial area in personalized medicine. The authors propose an innovative way to pre-train the model on drug synergy functions and use in-context learning for new, unseen drugs and cell lines. The significance of this work lies in its ability to move toward practical applications of AI in enhancing personalized therapy strategies, which could lead to more efficient and tailored treatment options for patients with complex diseases like cancer.

**Questions To Authors:**

- Data Preprocessing and Splits: Could you provide more detailed information on how the data was preprocessed and split for training, validation, and testing? Understanding this aspect is crucial for assessing the model's robustness and the potential for overfitting.
- Model Generalizability: How does SynerGPT perform when applied to other diseases or drugs not included in your study? Are there limitations in its applicability to other areas of drug discovery?
- Integration into Drug Discovery Pipelines: How do you envision integrating SynerGPT into real-world drug discovery pipelines? What are the challenges, and how might they be overcome? Additionally, how does the model handle the complexity of multi-drug interactions in more genetically diverse patient populations?

**Reasons To Accept:**

- Innovative Approach: The innovation of SynerGPT lies in its application of in-context learning, traditionally used in natural language processing, to the domain of drug synergy prediction. This approach demonstrates a novel use of GPT models outside of their usual textual contexts. As precision medicine continues to grow in importance, the ability to predict how different drugs interact at a personalized level becomes increasingly critical. This paper's focus on leveraging AI to predict personalized drug interactions makes it highly relevant to current medical research priorities.
- Strong Empirical Results: The paper details several experiments where SynerGPT matches or exceeds the performance of existing state-of-the-art models that use more complex, domain-specific inputs. These results suggest that the model is not only innovative but also practical and effective.
- Potential Impact: The methodology could significantly influence the field by providing a new tool for drug discovery and development. This impact is particularly noteworthy in the context of developing treatments for diseases with high variability between patients.

**Reasons To Reject:**

- Lack of Detail: The paper could be improved by providing more detailed technical descriptions of the model architecture and training processes. This would help in replicating the study and understanding the nuances of the model's performance.
- Limited Evaluation: The evaluation focuses primarily on specific types of cancer. Expanding the scope to include other diseases where drug synergy is critical could make the case stronger for the model's applicability and robustness.
- Concerns Over Generalizability: The paper does not sufficiently address how the model performs across different sets of drugs and diseases beyond those tested. This is crucial for understanding the model's utility in broader clinical settings.

---

> ### Author Rebuttal · Authors · 2024-05-31
>
> Thank you for your review of our paper, and for noting its “innovative approach”, “strong empirical results”, and “potential impact”.
>
> > Limited Evaluation:
> * We evaluate on cancer because there are sufficiently large, relevant datasets that have been created. This is because cell lines are frequently derived from cancer cells, whereas other diseases can be more difficult to create datasets for. We follow standard evaluation practices within this research area on datasets consisting of many types of cancer.
>
> > Lack of Detail:
> * Would you be able to specify which points in the technical description are unclear?
>
> > Concerns Over Generalizability:
> * While we cannot quantify performance on drugs/diseases that were not included in the dataset, we do create evaluation splits which exclusively contain drugs and cell lines unseen during the training process. Table 2 shows that in this setting we are able to improve model performance via in-context learning.
> * Since SynerGPT works via in-context learning, it can be adapted to other tasks fairly easily if they can be represented as a sequence.
>
> > Data Preprocessing:
> * The data is preprocessed following the methodology in Rozemberczki et al. 2022. The procedure for data splitting in section 6 is as follows: We select either $m$ drugs or cell lines in the “unknown drug” and “unknown cell line” setting, respectively. To construct our data split, we remove all “unknown” synergy tuples so that the remaining dataset only contains tuples with known drugs/cells (this is our training set $D^{Tr}$). The remaining unknown drug/cell tuples are randomly partitioned into three equal sized sets: a context bank $D^{c}$ , a validation set $D^{v}$ , and a test set $D^{Te}$.
>
> > Drug Discovery Pipelines:
> * This is an excellent question! We will add discussion about this into the revised paper along these lines:
>
> 1. Improving efficiency of drug synergy testing: In silico predictions can help reduce expensive real-world testing.
> 2. Designing new drugs for synergistic effects: Our method can help screen large datasets consisting of millions to billions of possible drug structures for possible synergies.
> 3. Patient-specific drug synergy prediction: One of the most interesting aspects of our methodology is predicting drug synergies specifically for a patient.
>
> * The cell lines in our dataset already represent genetic diversity in terms of the body tissue type, ethnicity, and biological sex of the originating organism (human).

---

> > ### Comment · Reviewer_FRwf · 2024-06-05
> >
> > Thank you for the response. These all make sense and I assume there will be (very minor) updates to the manuscript. My (positive) recommendation remains unchanged.

---

### Official Review · Reviewer_n3HL · 2024-05-11

**Rating:** 7
**Confidence:** 4
**Ethics Flag:** 1

**Summary:**

In-depth analysis of drug synergies is an important basis for cancer drug design and selection. Based on language models, this paper proposes a novel method for predicting drug synergies, namely in-context learning for synergy prediction. Without introducing many drug and cell line features, this work trains SynerGPT based on GPT-2, which can leverage the capabilities of the Transformer framework to achieve more competitive prediction performance than previous baseline models. This paper also uses algorithm to modify and optimize the context. Finally, this paper proposes the task of inverse drug design to discover medication regimens for specific patients. In short, this study cleverly leverages the capabilities of language models to provide a new, efficient, low-cost, and easily implementable solution for drug synergistic research.

**Questions To Authors:**

1. In Section 4, the paper concludes that “replacing drug and cell names with random tokens resulted in no drop in performance.” Does this conclusion in terms of Encoder-only models similarly apply to Section 5? Does the method of representing drug names with random tokens remain effective in the training of GPTs?
2. In Section 5, the inputs of SynerGPT, such as drugs and cell lines, are obtained from a learnable embedding layer. Compared to directly fine-tuning GPT-2 using real drug and cell line names as training data, how much difference in evaluation results will there be with the current SynerGPT?

**Reasons To Accept:**

1. This paper discovers that, in drug synergy prediction learning, drug and cell line features is not absolutely necessary in model training. An efficient method for drug synergy prediction using Transformer language models without drug features is explored. Based on a relatively smaller dataset, this approach utilizes randomized tokens to represent drugs and cell lines, which could reduce the reliance on large-scale datasets and high-quality knowledge bases. It helps lower the cost of drug synergy learning, presenting a meaningful contribution.
2. This work evaluates the performance of two types of models, BERTs and GPTs, in the task of drug synergy prediction through experiments, providing comprehensive and systematic experimental results and analysis. The trained model SynerGPT in this study demonstrates advantages compared to baseline models like DeepSynergy and DeepDDS, confirming the effectiveness of the approach. Additionally, the context of SynerGPT is also optimized..
3. A novel task of Inverse Synergistic Drug Structure Design is proposed, which utilizes GPTs to design corresponding drugs based on patient information. This is an interesting work that provides a new approach for personalized drug selection.

**Reasons To Reject:**

1. This paper fails to adequately consider the intrinsic features of drugs and cell lines. It may be overly simplistic to achieve drug and cell line representations solely through a learnable embedding layer. In current research, there are doubts about whether this method can be effective across a broader range of drugs and cell lines.
2. This paper only simply considers synergy between two drugs; no experiments are conducted regarding synergy among more drugs.

---

> ### Author Rebuttal · Authors · 2024-05-31
>
> Thank you for your thoughtful review. We appreciate your recognition of our paper’s contributions to demonstrating the “capabilities of language models to provide a new, efficient, low-cost, and easily implementable solution for drug synergistic research.”
>
> > This paper fails to adequately consider the intrinsic features of drugs and cell lines.[...] there are doubts about whether this method can be effective across a broader range of drugs and cell lines.
> * This is a good point. In Appendix Table 6, however, we do consider a version of the SynerGPT model architecture trained with features of drugs and cell lines (called “Features”), but we find that results are not improved. Further, in Appendix A we note that while we show that strong performance is possible without features, future work will still likely want to integrate external database features into drug synergy prediction; however, they will likely need to be integrated in a more thoughtful manner in order to ensure an actual benefit. Regardless, in our experimental results we show that our method can improve performance via in-context learning for unseen drugs and cell lines, so we believe it will still be effective.
>
> > This paper only simply considers synergy between two drugs; no experiments are conducted regarding synergy among more drugs.
> * This is an unfortunate limitation of existing datasets, but our method is easily extensible to n-tuple synergies when relevant datasets become available.
>
> > In Section 4, the paper concludes that “replacing drug and cell names with random tokens resulted in no drop in performance.” [...] the training of GPTs?
> * Yes, this is the case for Section 5. As shown in Table 2, a SynerGPT model trained with random tokens evaluated without in-context learning (“Zero-Shot SynerGPT”) has roughly the same performance as a GPT-2 model trained with names (“GPT-2”).
>
> > In Section 5, the inputs of SynerGPT, such as drugs and cell lines, are obtained from a learnable embedding layer. [...] how much difference in evaluation results will there be with the current SynerGPT?
> * We directly fine-tune GPT-2 using real drug and cell line names in the few-shot setting (“GPT-2” in Table 2). We find roughly 3% lower ROC-AUC than SynerGPT. Details are in Appendix B.2.
>
>
> Thank you again for your comments, which will help us strengthen the writing of our paper.

---

### Decision · Program_Chairs · 2024-07-10

**Decision:**

Accept

**Comment:**

The paper proposes a new approach for drug synergy prediction using fewer features/sources of pre-training knowledge compared to prior work. The results are interesting and surprising in the context of prior work.

It also defines an additional task to generate drug molecules that are synergistic in the context of a specific patient’s context and evaluates the language model on it.

This is an interesting application of Transformer models to in-context learning on sequences of drugs for a cell line.

Pros
* The paper presents interesting findings which can have implications for precision cancer medicine
* The additional experiments included after the author response addressed reviewers' comments

Cons
* The organization of the presentation is not optimal